# R-Capsule: Compressing High-Level Plans for Efficient Large Language Model Reasoning

## Abstract

Chain-of-Thought (CoT) prompting helps Large Language Models (LLMs) tackle complex reasoning by eliciting explicit step-by-step rationales. However, CoT's verbosity increases latency and memory usage and may propagate early errors across long chains. We propose the **Reasoning Capsule (R-Capsule)**, a framework that aims to combine the efficiency of latent reasoning with the transparency of explicit CoT. The core idea is to compress the high-level plan into a small set of learned latent tokens (a Reasoning Capsule) while keeping execution steps lightweight or explicit. This hybrid approach is inspired by the Information Bottleneck (IB) principle, where we encourage the capsule to be approximately minimal yet sufficient for the task. Minimality is encouraged via a low-capacity bottleneck, which helps improve efficiency. Sufficiency is encouraged via a dual objective: a primary task loss for answer accuracy and an auxiliary plan-reconstruction loss that encourages the capsule to faithfully represent the original textual plan. The reconstruction objective helps ground the latent space, thereby improving interpretability and reducing the use of uninformative shortcuts. Our framework strikes a balance between efficiency, accuracy, and interpretability, thereby reducing the visible token footprint of reasoning while maintaining or improving accuracy on complex benchmarks. Our codes are available at: https://anonymous.4open.science/r/Reasoning-Capsule-7BE0

## 1 Introduction

Large Language Models (LLMs) exhibit strong multi-step reasoning when prompted with Chain-of-Thought (CoT) Wei et al. (2022); Lightman et al. (2023). By instructing models to generate an explicit sequence of intermediate steps, CoT significantly improves performance on tasks ranging from arithmetic and commonsense reasoning to symbolic manipulation. However, explicit chains are costly: generating long sequences increases inference latency and memory usage. Furthermore, these long-form generations are susceptible to cascading errors, where a mistake in an early step compromises the entire reasoning process. As LLMs are increasingly deployed in latency- and cost-sensitive applications, the community has sought alternatives that preserve CoT's accuracy benefits while reducing its overhead.

Existing approaches to mitigate these issues can be grouped into three broad families, each with distinct trade-offs. Ensemble and sampling-based methods (e.g., self-consistency Wang et al. (2022); Yao et al. (2023a); Besta et al. (2024); Chen et al. (2025)) improve accuracy by aggregating multiple reasoning chains. While effective, they multiply inference-time cost and do not address verbosity at its root. Implicit or latent reasoning methods Deng et al. (2024); Hao et al. (2024) compress intermediate computation into dense vectors and decode short outputs, saving tokens. However, they are often opaque: compressing both planning and execution can entangle them, hindering verifiability and inviting shortcuts when the latent channel is unconstrained.

Hierarchical and modular reasoning approaches (e.g., plan-then-solve Huang et al. (2022); Wang et al. (2023)) separate high-level planning from low-level execution. This structure enhances faithfulness and controllability; however, plans are typically generated explicitly in natural language or via tool calls, reintroducing token overhead and sensitivity to exposure bias and plan-execution mismatch. These limitations motivate a more targeted question: can we obtain the efficiency of latent reasoning without sacrificing the structure and transparency of explicit plans? We answer this in the

affirmative by introducing the Reasoning Capsule. This framework compresses only the high-level plan into a compact, continuous latent while keeping execution lightweight and optionally explicit. Our key observation, supported by systematic experiments on arithmetic and commonsense reasoning benchmarks, is twofold: (1) explicitly generating textual plans before steps often degrades accuracy due to longer sequences and increased opportunities for error; yet (2) compressing the plan into a small number of contiguous latent tokens, while leaving execution uncompressed or lightly decoded, yields consistent gains over generating steps directly, and substantially outperforms compressing the execution itself. In other words, the plan is the right target for compression, whereas compressing the full CoT tends to discard useful inductive biases and supervision signals.

In the latent-planning stage, a decoder-only LLM projects its internal state through a low-capacity bottleneck to produce K capsule tokens. The model then conditions on these tokens (e.g., as soft prompts/prefix) for subsequent generation, replacing explicit plan text. In the execution stage, an auxiliary one-layer transformer (plan decoder) is trained to reconstruct the high-level plan and supervise CoT and answer generation from the capsule. At inference, we skip explicit CoT and directly generate the answer conditioned on the capsule; the auxiliary decoder is used only during training. This design respects the hierarchical nature of reasoning—planning versus execution—while minimizing the visibility of tokens. To make capsules compact and semantically meaningful, we adopt an IB-inspired design. Tishby et al. (2000). The bottleneck projection enforces minimality by constraining capacity. In contrast, a dual objective enforces sufficiency: a standard next-token loss for answer generation and an auxiliary reconstruction loss that trains the model to recover the high-level textual plan from the capsule. This reconstruction grounds the capsule in an interpretable strategy and counteracts latent collapse, avoiding uninformative shortcuts. Empirically, we find that (i) learning to first generate an explicit plan and then steps often harms performance; (ii) compressing the plan into latent tokens improves over step-first baselines; and (iii) further compressing the steps substantially degrades results—highlighting the asymmetry between plan and execution in what should be compressed. We validate our approach on arithmetic benchmarks and commonsense reasoning benchmarks. Across datasets, Reasoning Capsules deliver competitive or improved accuracy with fewer visible tokens and reduced latency compared to explicit CoT. They outperform entirely latent CoT schemes that compress both plan and steps. Ablations varying capsule length and bottleneck dimension illustrate a robust accuracy–efficiency trade-off. Qualitative analyses indicate that decoded plans remain faithful, and the model's attention is concentrated on capsule tokens during execution. Our contributions are threefold:

- We introduce Reasoning Capsules, a framework that compresses high-level plans into compact latent tokens to drive downstream execution, reconciling the efficiency of latent reasoning with the structure and interpretability of explicit plans.
- We provide a principled grounding via the Information Bottleneck, and instantiate it with an architectural bottleneck plus a plan-reconstruction objective that yields minimal yet sufficient, semantically grounded latents.
- We present a practical training recipe that integrates with GPT-based decoders and a lightweight one-layer decoder for supervision, along with comprehensive experiments on arithmetic and commonsense reasoning showing consistent token savings, latency reductions, and accuracy gains over explicit-plan and entirely latent CoT baselines.

## 2 METHODOLOGY

In this paper, we introduce **Reasoning Capsule**, a framework designed to enhance the efficiency and accuracy of multi-step reasoning in Large Language Models (LLMs). The core idea is to compress high-level strategic plans into compact, continuous latent representations. This approach mitigates the computational and statistical inefficiencies of generating verbose textual reasoning chains. We first present the overall framework, then provide a theoretical justification from the perspective of the Information Bottleneck principle, and finally detail the training objective.

### 2.1 FROM CHAIN-OF-THOUGHT TO LATENT PLANNING

Standard Chain-of-Thought (CoT) tackles a problem $Q$ by generating an explicit sequence of reasoning steps $S = (s_1, s_2, \ldots, s_N)$ before producing a final answer $A$. The whole generation process

is modeled as $p(S, A|Q)$. While effective, generating the explicit sequence $S$ token by token is computationally expensive and can introduce cascading errors.

Our key insight is that reasoning chains often exhibit a hierarchical structure: a high-level strategic plan (e.g., "first, calculate the discount; then, compute the final price") followed by low-level, step-by-step execution (e.g., "$\langle\langle 100 \times 0.2 = 20 \rangle\rangle$", "$\langle\langle 100 - 20 = 80 \rangle\rangle$"). We hypothesize that the high-level plan is a primary candidate for compression, as its semantic essence is more critical than its specific wording for guiding the final solution.

We therefore propose **Latent Planning**, a paradigm where the LLM first generates a compact latent representation of the high-level plan—the Reasoning Capsule—and then conditions on this capsule to execute the low-level reasoning steps. Let the explicit high-level plan be $P$ and the subsequent execution steps be $S_{\text{exec}}$. Our approach bifurcates the reasoning process:

**Latent Planning Stage:** Given $Q$, the model generates a set of capsules $C = \{c_1, \ldots, c_K\}$ that encode the high-level strategy originally articulated in $P$. This stage models $p(C|Q)$.

**Conditioned Execution Stage:** The model generates the execution steps $S_{\text{exec}}$ and the final answer $A$ conditioned on both the question $Q$ and the latent plan $C$. This stage models $p(S_{\text{exec}}, A|Q, C)$.

The overall generative process is thus factorized as $p(S_{\text{exec}}, A|Q, C)p(C|Q)$.

## 2.2 Architecture: Generating and Utilizing Reasoning Capsules

Our architecture is built upon a standard decoder-only transformer. We introduce a mechanism to generate and consume Reasoning Capsules within the forward pass (see Figure 1).

**Capsule Generation.** To generate a capsule, we prompt the model to emit a special '[CAPSULE]' token at the point where a textual plan would typically be articulated. The hidden state $h_t \in \mathbb{R}^D$ from the final transformer layer corresponding to this token is used as input to a bottleneck network. This network projects the high-dimensional hidden state into a low-dimensional capsule $c \in \mathbb{R}^d$, where $d \ll D$:

$$c = \text{Proj}(h_t) = W_p h_t + b_p, \tag{1}$$

where $W_p \in \mathbb{R}^{d \times D}$ and $b_p \in \mathbb{R}^d$ are learnable parameters. This projection acts as a structural implementation of the compression objective in the Information Bottleneck principle, forcing the model to distill the most salient strategic information from the context-rich hidden state $h_t$.

**Conditioning on Capsules.** Once generated, the capsule $c$ must guide subsequent reasoning. We project the capsule back into the model's input embedding space using a separate linear transformation. This projected embedding is then fed as input to the transformer at the beginning of the execution stage. This allows the capsule to condition the generation of all subsequent tokens ($S_{\text{exec}}$ and $A$), effectively acting as a compact, latent instruction that steers the model's computations.

## 2.3 Theoretical Grounding: The Information Bottleneck Perspective

A key challenge in latent variable models is ensuring the representations are both compressed and meaningful. Our design is formally motivated by the **Information Bottleneck (IB) principle** Tishby et al. (2000). The IB principle provides a framework for learning a compressed representation $Z$ of a source variable $X$ that is maximally informative about a target variable $Y$. The objective is to learn a mapping $p(Z|X)$ that maximizes the Lagrangian $\mathcal{L}_{\text{IB}} = I(Z; Y) - \beta I(X; Z)$, where $I(\cdot; \cdot)$ denotes mutual information and $\beta$ is a Lagrange multiplier. The source variable $X$ is the hidden state $h_t$, which contains rich, high-bandwidth information about the question $Q$ and reasoning context. The compressed representation $Z$ is the Reasoning Capsule $c$. The target variable $Y$ is the information required to solve the task, i.e., the execution steps and final answer ($S_{\text{exec}}, A$).

The goal is to learn a capsule $c$ that is a **minimal sufficient statistic** for the reasoning task.

**Minimality (Compression):** The capsule $c$ must be a compressed version of $h_t$. This corresponds to minimizing the mutual information $I(h_t; c)$, which forces the model to discard irrelevant information like specific phrasing or syntactic variations. Our bottleneck architecture (Eq. 1) directly serves

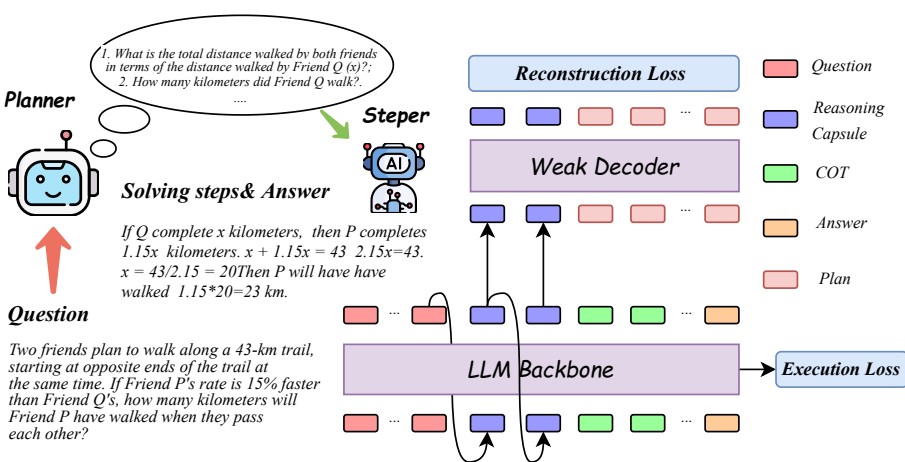

Figure 1: A conceptual diagram of our Reasoning Capsule framework. The LLM generates a compact latent capsule representing the high-level plan, which is passed through a bottleneck. This capsule conditions the subsequent generation of the execution steps and the final answer. An auxiliary reconstruction decoder ensures the capsule is semantically grounded by forcing it to reconstruct the original textual plan, guided by the Information Bottleneck principle.

this goal. By projecting $h_t \in \mathbb{R}^D$ into a low-dimensional space $c \in \mathbb{R}^d$ where $d \ll D$, we constrain the information capacity of the capsule, providing a strong inductive bias for compression.

**Sufficiency (Informativeness):** The capsule $c$ must retain all information from $h_t$ that is relevant for producing the correct solution by maximizing the mutual information $I(c; S_{\text{exec}}, A)$.

Directly optimizing $I(c; S_{\text{exec}}, A)$ is intractable. We instead use the original high-level textual plan, $P$, as an effective proxy. We hypothesize that $P$ encapsulates the core strategic information needed for the task. Therefore, we aim to maximize $I(c; P)$ as a surrogate objective for sufficiency. This ensures that the latent capsule is semantically grounded in the human-interpretable reasoning plan. Our training objective, detailed next, is a practical realization of this IB-based formulation.

## 2.4 GROUNDED TRAINING OBJECTIVE

To operationalize the IB principle, we train the model with a multi-task objective that balances task performance (sufficiency for the answer) and representational fidelity (sufficiency for the plan). The total loss $\mathcal{L}$ is a weighted sum of an execution loss and a plan reconstruction loss,

$$\mathcal{L} = \mathcal{L}_{\text{exec}} + \lambda \mathcal{L}_{\text{recon}}, \tag{2}$$

where $\lambda$ is a hyperparameter balancing the two objectives.

**Execution Loss ($\mathcal{L}_{\text{exec}}$).** This is the primary task loss, ensuring the capsule is sufficient for solving the problem. It is a standard auto-regressive cross-entropy loss for generating the target sequence $T = (S_{\text{exec}}, A)$, which includes both the intermediate execution steps and the final answer. The generation is conditioned on the question $Q$ and the generated set of capsules $C$:

$$\mathcal{L}_{\text{exec}} = -\log p(T|Q, C). \tag{3}$$

Minimizing this loss implicitly maximizes the mutual information $I(C; T)$, encouraging the capsules to be directly helpful for the downstream task.

**Reconstruction Loss ($\mathcal{L}_{\text{recon}}$).** This auxiliary loss serves as our practical method for maximizing $I(C; P)$, grounding the latent space and ensuring interpretability. We employ a separate, shallow transformer decoder that takes the sequence of capsules $C$ as input and is trained to reconstruct the original high-level textual plan $P$:

$$\mathcal{L}_{\text{recon}} = -\log p(P|C). \tag{4}$$

This loss forces each capsule to encode sufficient information to recover its corresponding textual plan component, ensuring the latent plan is a faithful and high-fidelity representation of the explicit one. This prevents the model from learning uninterpretable latent shortcuts, making the reasoning process more robust.

By combining these components, our framework learns to form compact, efficient, and semantically meaningful plans, thereby practically and effectively realizing the goals of the Information Bottleneck principle.

## 3 EXPERIMENTS

To validate the effectiveness and efficiency of our **Reasoning Capsule** framework, we conduct a comprehensive set of experiments on various reasoning benchmarks. Our evaluation is designed to answer several key research questions that stem from the claims made in our methodology.

- **RQ1 (Effectiveness):** Does our Reasoning Capsule framework outperform strong baselines, exceptionally standard Chain-of-Thought fine-tuning (CoT-SFT), in terms of reasoning accuracy?

- **RQ2 (Generalizability):** Is the performance improvement of Reasoning Capsules consistent across mathematical and commonsense reasoning domains?

- **RQ3 (Scalability):** How does the benefit of our latent planning approach scale with the size of the base language model?

- **RQ4 (Efficiency):** Does the latent planning paradigm lead to a more compact and efficient reasoning process, measured by generation length and latency?

- **RQ5 (Interpretability):** Do the latent capsules encode genuine, verifiable planning information?

### 3.1 EXPERIMENTAL SETUP

#### 3.1.1 DATASETS

We evaluate our method on standard benchmarks for mathematical and commonsense reasoning.

- **Mathematical Reasoning**: **GSM8K** Cobbe et al. (2021) (grade-school math word problems), **MultiArith** Roy & Roth (2016) (math problems requiring multiple reasoning steps), and **AQuA** Ling et al. (2017) (multiple-choice algebraic word problems).

- **Commonsense Reasoning**: **StrategyQA** Geva et al. (2021) (yes/no questions requiring a multi-step reasoning strategy) and **CommonsenseQA 2.0 (CSQA2)** Talmor et al. (2018; 2022) (a challenging multiple-choice QA dataset requiring prior knowledge).

#### 3.1.2 CoT DATA GENERATION

We generate the CoT data using the `gpt-o3` model via few-shot prompting. For each problem, we prompt the model to produce a solution plan and step-by-step reasoning. During this process, the final answer is withheld from the model. We employ rollout sampling with a temperature of 1.0 and repeat the generation up to 10 times per problem, stopping once a process yielding the correct answer is found. If all 10 attempts fail to produce a proper solution, we then provide the model with the correct answer and prompt it to generate a valid reasoning path, including the plan and steps. The generated prompts and cases are provided in Appendix B.

#### 3.1.3 BASE MODELS

To test the scalability (RQ3), we conduct experiments on three decoder-only transformer models of varying sizes: **GPT-2 (0.2B)** Radford et al. (2019), **LLaMA-3 (1B)** Dubey et al. (2024), **LLaMA-3.1 (7B)** Dubey et al. (2024), **Qwen-3 (8B)** Yang et al. (2025). All methods are fine-tuned on the same pre-trained checkpoints for a fair comparison.

Table 1: Main results on mathematical and commonsense reasoning benchmarks. We report accuracy (%) on five datasets for methods applied to GPT-2 (115M) and LLaMA-3 (1B) models. Our R-Capsule consistently outperforms the strong CoT-SFT baseline.

| Method | Mathematical Reasoning | | | Commonsense Reasoning | |
|---|---|---|---|---|---|
| | GSM8K | MultiArith | AQuA | StrategyQA | CSQA2 |
| *Model: GPT-2 (150M)* | | | | | |
| Standard SFT | 19.1 | 78.5 | 28.1 | – | – |
| CoT-SFT | 42.9 | 86.9 | 33.2 | – | – |
| Plan-SFT | 37.5 | 82.6 | 30.9 | – | – |
| Coconut | 34.1 | 84.8 | 32.8 | – | – |
| iCoT | 41.5 | 85.2 | 33.0 | – | – |
| **R-Capsule (Ours)** | **46.2** | **92.4** | **37.9** | – | – |
| *Model: LLaMA-3 (1B)* | | | | | |
| Standard SFT | 44.1 | 89.0 | 35.5 | 60.5 | 55.1 |
| CoT-SFT | 59.7 | 94.1 | 48.4 | 62.9 | 57.2 |
| Plan-SFT | 59.7 | 94.1 | 44.8 | 63.4 | 56.5 |
| **R-Capsule (Ours)** | **63.8** | **96.5** | **52.1** | **66.8** | **59.8** |

### 3.1.4 Baselines

We compare our **Reasoning Capsule** framework against a series of strong baselines:

**Standard SFT (w/o CoT):** A standard supervised fine-tuning baseline where the model is trained to directly predict the final answer A from the question Q, that is, modeling p (AQ). This establishes the performance without any explicit reasoning steps.

**CoT-SFT:** The standard Chain-of-Thought fine-tuning approach Wei et al. (2022). The model is trained to generate the complete textual reasoning chain S followed by the final answer A, modeling p (S, AQ). This is our main and strongest baseline.

**Coconut:** A method that improves reasoning by generating multiple reasoning paths and using a verifier to select the most consistent one, thereby enhancing robustness Hao et al. (2024).

**iCoT:** a method that allows language models to gradually internalize chain-of-thought (CoT) reasoning steps by incrementally removing intermediate CoT tokens and fine-tuning, thereby achieving implicit CoT reasoning with high accuracy and fast inferenceDeng et al. (2024).

**Plan-SFT:** a unified post-training framework that distills synthetic "planning trajectories" (task decompositions) from large-scale LLMs and fine-tunes smaller open-source LLMs via supervised fine-tuningParmar et al. (2025).

### 3.1.5 Implementation Details

We employ the AdamW optimizer with a learning rate of $5 \times 10^{-6}$, where $\beta_1$ and $\beta_2$ are set to 0.9 and 0.999, respectively, and the weight decay is 0.01. The learning rate follows a cosine schedule with a linear warmup over the first 10% of total training steps. We use a total batch size of 32 (4 per GPU across 8 NVIDIA A800 GPUs) without gradient accumulation. The training epochs are set differently for various models: 5 epochs for GPT2, and 3 epochs for LLaMA3-1B, LLaMA3-7B, and Qwen3-8B. The length of the Reasoning Capsule is fixed at 2. For the loss function, the reconstruction loss (adopting MSE loss) is weighted by $\lambda = 0.5$ against the main task loss.

### 3.2 Main Results (RQ1 & RQ2)

The results in Table 1 demonstrate the effectiveness and generalizability of our Reasoning Capsule framework (RQ1 and RQ2). Across both GPT-2 and LLaMA-3 (1B) backbones, our method consistently surpasses the strong CoT-SFT baseline on all five mathematical and commonsense reasoning benchmarks. For instance, on GSM8K, R-Capsule provides a +3.3% and +4.1% absolute improvement for GPT-2 and LLaMA-3, respectively. This confirms the core benefits of our latent planning approach on established model sizes.

## 3.3 SCALABILITY ANALYSIS (RQ3)

To address our third research question (RQ3) concerning the scalability of our approach, we conducted a focused evaluation on two contemporary 7-billion-parameter models: `LLaMA-3-7B` and `Qwen3-8B`. In this analysis, we compare our **R-Capsule** against the **CoT-SFT** baseline on the GSM8K (mathematical reasoning) and StrategyQA (commonsense reasoning) benchmarks. This enables us to evaluate whether the performance gains of our method scale effectively with model size across various reasoning domains.

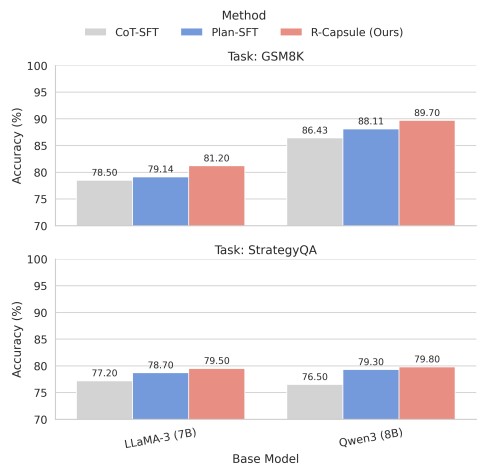

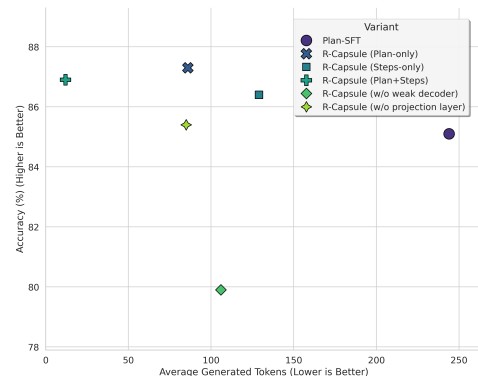

(a) Performance of models with increasing parameter counts. R-Capsule maintains and, in some cases, enhances its accuracy gains at larger scales.

(b) Results of ablation studies. The contributions of different components of R-Capsule are analyzed, showing their impact on overall performance.

Figure 2: (a) Scalability analysis: Accuracy (%) on a representative task with increasing model size. (b) Ablation study: Effects of removing individual components on model performance.

The results, presented in Figure 2a, demonstrate that the advantages of the R-Capsule framework persist and are even amplified on these larger models. For instance, on `LLaMA-3-7B`, R-Capsule achieves a significant absolute improvement of **2.7%** on GSM8K and **2.3%** on StrategyQA over the CoT-SFT baseline. A similar trend is observed on `Qwen3-8B`, where our method yields a notable improvement of **3.27%** on GSM8K and **3.3%** on StrategyQA. These consistent gains across two distinct and powerful foundation models strongly suggest that the structural benefits of latent planning, as embodied by our R-Capsule, represent a general principle that scales effectively with model capability. This finding provides a robust affirmative answer to RQ3.

## 3.4 ABLATION STUDY: WHERE TO COMPRESS

We ablate which part of the reasoning process to compress. We augment the reasoning chain with an explicit textual **plan**, creating two components: the plan and the steps. We test four variants:

- **Plan-SFT**: Explicit plan → explicit steps (no compression).
- **R-Capsule (Plan-only)**: Latent plan → explicit steps. This is our main proposal.
- **R-Capsule (Steps-only)**: Latent steps, analogous to implicit CoT.
- **R-Capsule (Plan+Steps)**: Latent plan → latent steps (maximal compression).

Figure 2b shows that compressing only the plan (**R-Capsule (Plan-only)**) achieves the best accuracy-efficiency trade-off. It improves accuracy over the explicit `Plan-SFT` baseline while reducing generated tokens by over 60% (e.g., from 244 to 86 on Qwen3 8B). This suggests that encoding the high-level plan latently provides a robust guide for generating explicit, low-level steps. In contrast, compressing the detailed steps (`Steps-only` or `Plan+Steps`) is less effective, with the latter showing a drop in accuracy despite achieving maximum compression. Finally, ablating

Table 2: Token budget and latency comparison on GSM8K (Qwen3). Latency is measured with a batch size of 1 on the A100.

| Method | Tokens to Answer | Compression Ratio | Latency (s) |
| --- | --- | --- | --- |
| Explicit Plan-CoT | 447 | 1.0 | 3.12 |
| R-Capsule (Plan-Latent) | **232** | **0.52** | **1.47** |

the **projection layer** or the **weak decoder** leads to significant performance degradation (e.g., -7.4% acc. on Qwen3 without the weak decoder), confirming their architectural importance.

## 3.5 EFFICIENCY AND LENGTH ANALYSIS (RQ4)

To address our fourth research question (RQ4) concerning efficiency, we analyze the generation length and inference latency of our proposed method. As detailed in Table 2, we evaluate our R-Caps (Plan-Latent) approach against the Explicit Plan-CoT baseline on the GSM8K benchmark. We measure two key metrics: the total number of tokens generated to reach the final answer and the end-to-end inference latency on a single A100 GPU. The results demonstrate a substantial improvement in efficiency. Our R-Capsule (Plan-Latent) method requires only 232 tokens to derive an answer, marking a 48% reduction compared to the 447 tokens used by the baseline. This corresponds to a compression ratio of 0.52, indicating that our method can produce solutions that are nearly half the length of standard explicit reasoning chains. This significant reduction in token generation directly translates to a notable decrease in latency. The inference time drops from 3.12 seconds for Explicit Plan-CoT to just 1.47 seconds for our method, achieving a 2.12x speedup. This efficiency gain stems from our model's ability to operate on compact latent representations of the plan, bypassing the need to generate verbose, token-intensive intermediate steps.

## 3.6 INTERPRETABILITY OF LATENT CAPSULES (RQ5)

To validate our core hypothesis, that the bottleneck architecture distills a plan into a compact, abstract latent representation, we investigate the information encoded within these latent capsules. By analyzing the output of a weak decoder fed with a corresponding capsule, we aim to provide qualitative evidence that these capsules preserve the plan's essential logical structure, effectively functioning as abstract 'latent thoughts' rather than mere textual compressions. As the case demonstrates, the output from the weak decoder (weak decoder plan) is significantly more concise than the original plan. It strips away redundant descriptive language, distilling the core steps into direct. This provides strong evidence for our central hypothesis: the bottleneck architecture incentivizes the model not merely to compress the plan, but to compile it into an abstract computational graph.

> **Case Study: Model Output**
>
> "question": "In a spelling contest, Peter and Christina are on one team... Peter misses seven words and Christina misses 6, fewer than half the words Peter spelled correctly. How many words were misspelled by their team?"
> "plan": "Here's a plan to solve the problem: 1. Calculate Peter's correct words... 2. Find half of Peter's correct words... 3. Determine Christina's incorrect words... 4. Sum Peter's and Christina's incorrect words..."
> "weak decoder plan": "Determine the number of words Peter spelled correctly. Calculate half of the words Peter spelled correctly. Determine the number of words Christina spelled incorrectly. Calculate the total number of words misspelled by the team."
> "steps": "1. Peter's correct words: $50 - 7 = 43$. ... 4. Total team incorrect words: $7 + 15 = 22$.

Further analysis (Appendix C) supports these findings:

- **Vocabulary Distribution:** Projecting latent tokens into the vocabulary space reveals a focus on abstract verbs (e.g., 'calculate', 'total') rather than specific numbers, indicating they capture high-level intent.

- **Attention Analysis:** The decoder attends heavily to the latent plan tokens when generating subsequent calculation steps, confirming they serve as a guide.

This evidence confirms that latent capsules are most effective for representing high-level strategic plans, while explicit generation remains crucial for detailed, procedural steps.

## 4 RELATED WORK

Chain-of-Thought Prompting Chain-of-Thought (CoT) promptingWei et al. (2022) has been shown to significantly enhance performance on complex tasks by generating explicit, step-by-step reasoning tracesSun et al. (2024). Variants like self-consistency aggregationWang et al. (2022) further improve reliability by ensembling multiple reasoning chains. Representative examples of this paradigm include o1El-Kishky (2024) and DeepSeek R1DeepSeek-AI et al. (2025)—both reasoning models that have achieved strong performance. Collectively, these findings confirm that CoT prompting effectively addresses the challenges of complex task reasoning: its explicit step-by-step trace design breaks down intricate problems into manageable logical segments, while variant optimizations (e.g., tree of thoughtsYao et al. (2023b),self-refineYang et al. (2024)) mitigate uncertainty in reasoning processes. The strong performance of models such as o1 and DeepSeek R1 further validates that the CoT paradigm is not only theoretically sound but also practically impactful, becoming a foundational approach for enhancing reasoning capabilities in advanced models.

Latent Planning and the Information Bottleneck However, the verbosity of CoT not only increases inference latency and memory overheadHong et al. (2025) but also risks propagating errors across long reasoning sequences. Implicit or latent reasoning schemesDeng et al. (2023); Ye et al. (2025) compress intermediate reasoning steps into dense vectorsHao et al. (2024), enabling faster inferenceCheng & Durme (2024) but at the cost of reduced interpretability. Without explicit grounding, such representations may encode spurious shortcuts. By contrast, we leverage the Information Bottleneck principleTishby et al. (2000) to achieve two key goals: (i) enforcing minimality via a low-dimensional bottleneck, and (ii) ensuring sufficiency through an auxiliary reconstruction loss that recovers the original high-level plan. This dual objective guarantees that each *Reasoning Capsule* is both compact and semantically faithful. For additional details or supplementary materials about the content discussed in this section/chapter, readers are kindly referred to Appendix D.

Hierarchical and Modular Reasoning Hierarchical reasoning frameworks decouple planning from execution, e.g., in plan-and-solve prompting or modular CoT Parmar et al. (2025). Tool-oriented methods (e.g., ToolformerSchick et al. (2023), ReActYao et al. (2023c)) similarly structure the reasoning process into tool selection and execution. These approaches, however, rely on generating and parsing explicit plans or tool invocation commandsHao et al. (2023), which introduces extra computational overheadWang et al. (2024). Our method internalizes high-level planning within a latent space, eliminating the need for explicit plan text or tool-specific syntax. A subsequent reconstruction module then verifies that this latent plan accurately encapsulates the intended reasoning strategy, thereby unifying structural rigor and inference efficiency within a single model.

## 5 CONCLUSION

In this paper, we introduce Reasoning Capsules (**R-Capsule**), a hybrid framework that reconciles the efficiency of latent reasoning with the transparency of explicit CoT. Our key insight is to decouple high-level planning from low-level execution, compressing only the strategic plan into a compact set of latent tokens—the capsule. Grounded in the Information Bottleneck principle, our method enforces the capsule to be both minimal, by discarding redundant information via a low-capacity bottleneck, and sufficient, by optimizing a dual objective for task accuracy and plan reconstruction. This design preserves the high-level reasoning structure while drastically reducing token overhead. Extensive experiments on mathematical (e.g., GSM8K) and commonsense (e.g., StrategyQA) reasoning benchmarks demonstrate that R-Capsule significantly outperforms strong baselines across various model sizes. Ablation and interpretability studies confirm that our approach of compressing only the plan yields an optimal trade-off and that the capsules encode meaningful strategic intent. R-Capsule establishes that targeted compression of high-level plans is a principled and effective path toward efficient, accurate, and interpretable LLM reasoning.

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

## A USE OF LARGE LANGUAGE MODELS

We utilized Large Language Models (LLMs), such as Gemini-2.5-Pro, during the preparation of this manuscript. The usage was twofold: 1) for polishing the language, which included correcting grammatical errors and improving sentence clarity; and 2) as a brainstorming partner to discuss and refine technical details. The authors retained complete control over the content, and all final ideas, claims, and text are our own. We take full responsibility for the entire paper.

## B CHAIN-OF-THOUGHT DATA GENERATION DETAILS

To ensure the reproducibility of our Chain-of-Thought (CoT) training data, this section provides the complete few-shot prompt template used for generating high-level plans and step-by-step execution steps via GPT-o3, along with representative examples of the generated outputs.

### B.1 FEW-SHOT PROMPT TEMPLATE FOR COT GENERATION

The prompt template follows a structured format to guide GPT-o3 in distinguishing between high-level plans (task decomposition logic without specific calculations) and low-level execution steps (concrete mathematical operations or reasoning details). The template includes explicit instructions and two reference examples, as shown below:

> **Prompt**
>
> Provide a question you need to generate a problem-solving idea based on the question, solution and answer to split the problem into multiple sub-questions as a problem-solving plan and the mathematical formula steps. Here I give two examples. example 1. xxx example 2. xxx As shown in the example, you need to output the "plan", "steps" and "answer" sections in the format of "plan: xxx steps:xxx answer: xxx". Do not output other contents.

### B.2 REPRESENTATIVE GENERATED CoT EXAMPLES

We provide two complete examples of CoT outputs generated using the above template, corresponding to a mathematical reasoning problem (from GSM8K) and a commonsense reasoning problem (from StrategyQA), respectively. These examples illustrate the consistent distinction between "plan" (task decomposition) and "steps" (concrete execution) in our data.

#### B.2.1 EXAMPLE FROM GSM8K (MATHEMATICAL REASONING)

> **Case Study: Model Output**
>
> Problem: Two friends plan to walk along a 43-km trail, starting at opposite ends of the trail at the same time. If Friend P's rate is 15% faster than Friend Q's, how many kilometers will Friend P have walked when they pass each other?
> Generated Output:Plan: 1. Define a variable to represent the distance walked by Friend Q when they meet. 2. Express Friend P's distance in terms of Q's distance using the given speed ratio (since time is equal for both, distance ratio equals speed ratio). 3. Set up an equation based on the total trail length (sum of P's and Q's distances = 43 km). 4. Solve the equation to find Q's distance, then calculate P's distance.
> Steps: 1. Let $x$= distance (in km) walked by Friend Q when they meet. 2. Friend P's speed is 15% faster than Q's, so P's distance =$(1.15x)$. 3. Total trail length:$(x + 1.15x = 43)km$. 4. Combine like terms: $(2.15x = 43)$. 5. Solve for$x : (x = 432.15 = 20)km$ (Q's distance). 6. Calculate P's distance:$(1.1520 = 23)km$.
> Answer: 23

#### B.2.2 EXAMPLE FROM STRATEGYQA (COMMONSENSE REASONING)

> **Case Study: Model Output**
>
> Problem: Can a penguin fly from the South Pole to the North Pole?
> Generated Output: Plan: 1. Confirm the key biological characteristic of penguins related to flight. 2. Analyze the geographical distance between the South Pole and North Pole, and the required mode of travel for such a distance. 3. Evaluate whether penguins' physical capabilities match the travel requirements. 4. Conclude whether the journey is possible.
> Steps: 1. Penguins are flightless birds—they have wings adapted for swimming (flippers) but lack the wing structure and muscle strength needed for flight. 2. The distance between the South Pole and North Pole is approximately 12,742 km (diameter of Earth), which would require sustained flight for days (even for flying birds). 3. Since penguins cannot fly, they cannot traverse this distance via air; additionally, ocean routes between the poles are blocked by ice sheets, making swimming impossible. 4. Thus, a penguin cannot fly from the South Pole to the North Pole.
> Answer: No

## C INTERPRETABILITY

To understand why this implicit plan representation is so effective, we conducted an in-depth analysis of the information encoded within these latent tokens. We set the number of latent plan tokens

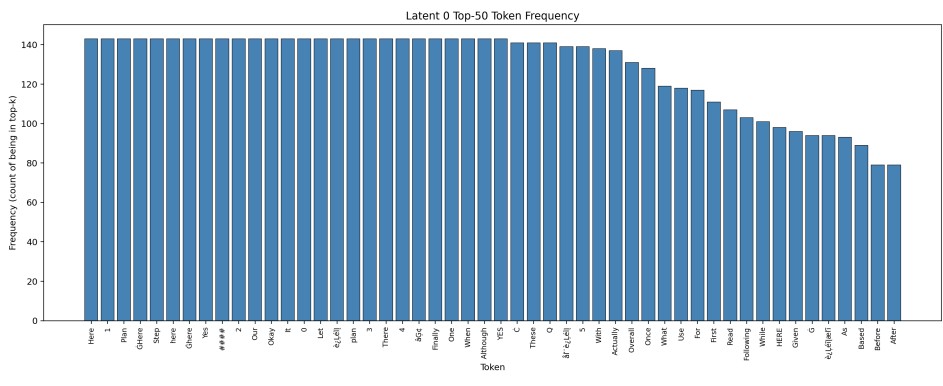

(a) First latent plan token.

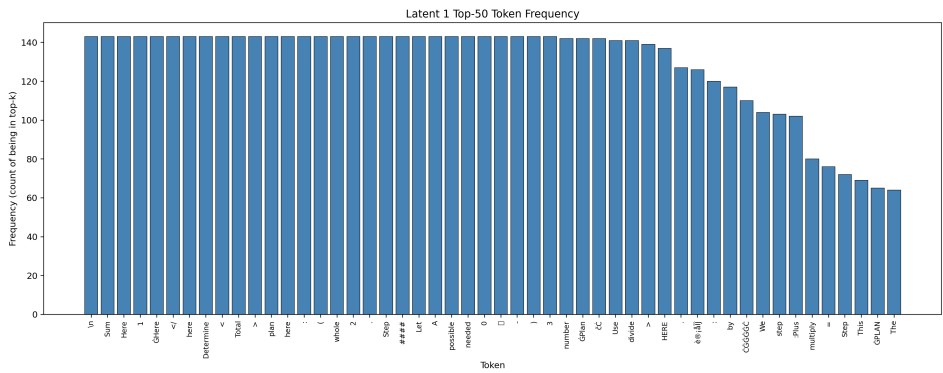

(b) Second latent plan token.

Figure 3: Vocabulary distribution of the LM head logits for the two latent plan tokens, aggregated over 200 test cases. The analysis reveals an explicit functional specialization. **(a)** The first token learns to encode **intent and initiation**, with its top predictions dominated by discourse markers ('Here', 'Plan', 'First') and instructional verbs ('Calculate', 'Find'). **(b)** The second token focuses on **structure and execution**, predicting structural elements (newlines, parentheses), mathematical operators ('-', '*'), and operational terms ('Sum', 'multiply') to format the subsequent step.

to two. We analyzed the vocabulary distribution of the language model head's logits corresponding to each token across a sample of 200 different problems. The results, visualized in Figure 3, reveal a fascinating specialization of roles between the latent tokens.

**The First Latent Plan Token: Encoding Intent and Initiation.** As shown in Figure 3(a), the vocabulary distribution for the first latent token is dominated by high-frequency "discourse markers" and "initiator" words. Tokens such as Here, Plan, Step, Let's, and First appear with high probability. This suggests that the first token has learned to function as a structural signal, activating a "planning" or "reasoning-initiation" mode within the model. Furthermore, the presence of instructional verbs like Calculate, Think, Find, and Determine in the mid-frequency range indicates that this token also captures the high-level intent or the primary cognitive action required for the initial part of the plan. It essentially tells the model, "Begin reasoning, and the first major goal is to calculate/find something."

**The Second Latent Plan Token: Encoding Structure and Execution.** The distribution for the second latent token, shown in Figure 3(b), paints a different but complementary picture. This to-ken's top predictions are heavily skewed towards structural and mathematical symbols, including newline characters (\n), comparison operators (¡, ¿), parentheses ((, )), and arithmetic operators (-, *, =). This strongly indicates that the second token has specialized in encoding the executional and structural format of the subsequent calculation step. It prepares the model for the precise symbolic manipulation required, acting as a bridge between the high-level intent (from the first token) and

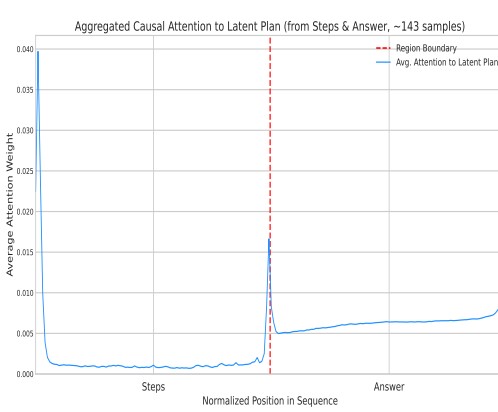

(a) Aggregated causal attention weights across 143 test samples

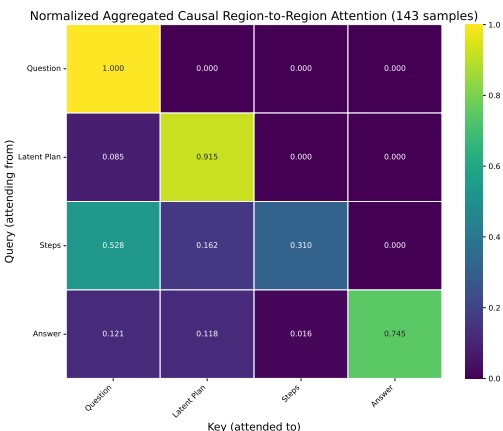

(b) Normalized average attention weight distribution

Figure 4: Hierarchical attention analysis of Reasoning Capsules.

the low-level, formatted output of the CoT step. The co-occurrence of tokens like Sum, Total, number, divide, and multiply further solidifies its role in priming the model for specific mathematical operations.

## C.1 ATTENTION ANALYSIS

To validate the Reasoning Capsule's role in guiding generation, we quantitatively analyzed the decoder's attention mechanism. Using 143 samples from the GSM8K dataset, we tracked the attention from the generated `Steps & Answer` sequence to the `Latent Plan` (Reasoning Capsule).

For the analysis, we segmented the input into three regions: `Question`, `Latent Plan`, and `Steps & Answer`. We then calculated the average attention from the `Steps & Answer` region to the `Latent Plan` region, aggregated across all samples. The results are visualized as:

- **Attention Curve (Figure 4a):** Plots the average attention weight against the normalized position in the `Steps & Answer` sequence, showing the attention trend during generation.
- **Inter-Region Attention Heatmap (Figure 4b):** An aggregated heatmap showing the attention flow between all defined regions (Query and Key).

## C.2 ATTENTION CURVE (FIGURE 4A): SUSTAINED GUIDANCE

The attention to the Latent Plan remains high and stable throughout the generations, confirming its continuous guiding role.

- **Step Generation (Normalized Position 0–0.8):** Attention is stable, indicating that the model continuously references the high-level plan while generating low-level calculation steps.
- **Answer Generation (Normalized Position 0.8–1.0):** Attention slightly increases as the model cross-references the plan to ensure the final answer aligns with the initial strategy.

## C.3 INTER-REGION ATTENTION HEATMAP (FIGURE 4B): LATENT PLAN DOMINANCE

The heatmap quantifies the Latent Plan's dominance in attention allocation.

- **Strong Guidance for Generation:** The attention from *Steps* to the *Latent Plan* (0.745) and from *Answer* to the *Latent Plan* (0.310) is significantly higher than to the original *Question* (0.121 and 0.162, respectively). This confirms the capsule acts as the primary strategic guide.

- **Information Compression:** Low attention to the *Question* suggests the Latent Plan effectively extracts and condenses all necessary information, making repeated access to the original problem unnecessary.
- **Reduced Error Propagation:** Near-zero self-attention within the generated steps (*Steps* → *Steps*) indicates the model relies on the global Latent Plan as a unified reference rather than on preceding steps, which helps mitigate cascading errors.

# D  INFORMATION BOTTLENECK FORMULATION FOR REASONING CAPSULES

We formalize the training objective of Reasoning Capsules under the **Information Bottleneck (IB)** principle (Tishby et al., 2000). Our goal is to learn a compressed latent representation $C \in \mathbb{R}^d$ of the high-level plan that is *minimal* yet *sufficient* for both reconstructing the original plan $P$ and predicting the final answer $A$.

## D.1  IB OBJECTIVE

Given the hidden state $\mathbf{h}_t \in \mathbb{R}^D$ at the capsule-token position, we seek a stochastic encoding $p(C \mid \mathbf{h}_t)$ that solves

$$\min_{p(C|\mathbf{h}_t)} \underbrace{I(\mathbf{h}_t; C)}_{\text{compression}} - \beta \underbrace{I(C; P)}_{\text{plan reconstruction}} - \gamma \underbrace{I(C; A)}_{\text{answer prediction}} \tag{5}$$

where

- $I(\mathbf{h}_t; C)$ enforces **minimality** by limiting the information contained in the capsule;
- $I(C; P)$ ensures **sufficiency** for reconstructing the textual plan $P$;
- $I(C; A)$ guarantees that the capsule is predictive of the final answer $A$;
- $\beta, \gamma > 0$ are Lagrange multipliers controlling the trade-off between compression and sufficiency.

## D.2  PARAMETRIC APPROXIMATION

In practice, we employ a **deterministic encoder** with a linear bottleneck

$$C = \text{Proj}(\mathbf{h}_t) = \mathbf{W}_p \mathbf{h}_t + \mathbf{b}_p, \qquad \mathbf{W}_p \in \mathbb{R}^{d \times D}, \ d \ll D. \tag{6}$$

This structural bottleneck enforces hard minimality: under a Gaussian assumption, the mutual information is upper-bounded by $I(\mathbf{h}_t; C) \leq \frac{d}{2} \log(2\pi e \sigma^2)$.

## D.3  TRAINING OBJECTIVE AS IB SURROGATE

We optimize a variational upper bound on Eq. equation 5:

$$\mathcal{L}_{\text{IB}} = \underbrace{-\log p_\theta(P \mid C)}_{\text{plan reconstruction}} + \lambda \underbrace{-\log p_\phi(A \mid C, Q)}_{\text{answer prediction}} \tag{7}$$

where

- $p_\theta(P \mid C)$ is a *shallow* transformer decoder that regenerates the plan;
- $p_\phi(A \mid C, Q)$ is the primary model that produces the final answer;
- $\lambda$ balances the two losses, and is numerically equivalent to the ratio $\beta/\gamma$ in Eq. equation 5.

# E  LATENT TOKEN NUMBER ABLATION

To validate the choice of latent token number $K$ (fixed as 2 in the main text), we conduct ablation experiments on $K \in \{1, 2, 3, 4\}$ using GSM8K (mathematical reasoning) and StrategyQA

Table 3: Latent Token Number $K$ Ablation on Qwen3-8B

| Model | $K$ | GSM8K Acc. (%) | GSM8K Tokens | StrategyQA Acc. (%) | Latency (s) |
|---|---|---|---|---|---|
| | 1 | 87.9 | 82 | 78.1 | 1.53 |
| | 2 | 89.7 | 86 | 79.8 | 1.65 |
| Qwen3-8B | 3 | 86.2 | 103 | 79.5 | 1.89 |
| | 4 | 85.5 | 118 | 78.9 | 2.11 |

(commonsense reasoning) datasets. We evaluate accuracy, generated tokens (before answer), and inference latency (batch size=1 on A100), with results shown in Table 3.

Key observations: 1. $K = 2$ achieves the highest accuracy across models/datasets. $K = 1$ under-represents the plan (lower accuracy), while $K \geq 3$ increases token count/latency without accuracy gains (redundant information). 2. Latency grows linearly with $K$, as more latent tokens require additional projection/computation. Thus, $K = 2$ balances accuracy and efficiency.

