# OpenReview forum: "R-Capsule: Compressing High-Level Plans for Efficient Large Language Model Reasoning"
_ICLR.cc/2026/Conference — ICLR 2026 Conference Withdrawn Submission_

### Official Review · Reviewer_efRb · 2025-10-26

**Soundness:** 2
**Presentation:** 2
**Contribution:** 2
**Rating:** 0
**Confidence:** 4

**Summary:**

The paper proposes a new method for LLM reasoning, R-Capsule, where LLMs first output high-level plans which are in a latent space and then textual detailed steps and finally the answer. The authors choose several benchmarks on math reasoning (such as GSM-8k) and commensense reasoning (such as strategyQA). They tested four models: GPT-2 (0.2B), LLaMA-3 (1B), LLaMA3.1 (7B)  and Qwen-3 (8B) and compared with baselines such as standard SFT, SFT+CoT, iCoT, coconut, etc.

**Strengths:**

1. The questions studied is important, i.e., improving LLM reasoning capability while reducing inference cost and increasing the interpretability.
2. The idea of using a high-level plan in latent space before the detailed execution steps makes sense.

**Weaknesses:**

1. I think the experiments are not comprehensive to make solid conclusions. The author chooses several benchmarks, four models and several methods. However, for each research question (RQ), they only choose a subset of benchmarks, models or methods to claim their method is better, while for difference RQs they use different subsets. This is not convincing to prove the effectiveness of the proposed method due to a “selective bias”.
2. The chosen benchmark are kind of “obsolete” and not challenging enough. For example, for math reasoning, the authors might consider test the proposed method in MATH or AIME, which are widely used for many current research.
3. I believe the authors miss an important previous work [1]. The previous work [1] proposed a method that is very similar to this paper (the major difference might be in their paper, the high-level tokens are in a discrete latent space while in this paper the high-level planning capsules are in a continuous latent space). However, the whole ideas are quite similar. I think it is important to discuss the differences between the two methods and include [1] as in important baseline to show the proposed method is better.
4. For RQ4, I don’t think it is fair to compare R-capsule with Plan-SFT. In the main table, Plan-SFT performs worse than CoT-SFT and the authors also claimed that CoT-SFT is the strongest baseline. In that case, the authors should compare generation length and latency with CoT-SFT to claim the proposed method is efficient.
5. The format looks a bit messy. For example, for citation, many \citet should be \citep and they are not separated from the main text, which makes reading difficult. In the related work section, the title of each paragraph should be bolded or should have a period.

**Reference:**

[1] Su, DiJia, Hanlin Zhu, Yingchen Xu, Jiantao Jiao, Yuandong Tian, and Qinqing Zheng. "Token Assorted: Mixing Latent and Text Tokens for Improved Language Model Reasoning." In Forty-second International Conference on Machine Learning.

**Questions:**

1. What’s the architecture of the weak decoder in Figure 1?
2. Why does table 1 only contain the results of two small models? Also, why it does not have the result of GPT-2 for commonsense reasoning? It also does not contain the result of coconut and iCoT for 1B model.  Since for the two small models, the most standard CoT+SFT achieves better performance than other methods (such as iCoT, coconut, etc.), does that mean the models chosen in the table are too small such that other methods do not show advantage? Or in other words, does that mean R-capsule can only outperform other methods in the regime of <= 150M models?
3. I’m not sure whether the claim in Section 3.4 that R-capsule (Plan-only) achieves the best accuracy-efficiency is very convincing. According to Figure 2(b), the accuracy difference between R-capsule (plan-only) and R-capsule (Plan+steps) is quite small and even negligible, while their efficiency differs a lot. Also, what’s the task tested in Figure 2(b)?
4. Does the weak decoder aim to reproduce the plan word by word or only keep its semantics? It seems you’re calculating a reconstruction MSE loss so you need to reproduce it exactly, but in section 3.6 you also mentioned that the weak decoder will output a more concise plan. Am I missing anything?
5. Does the weak decoder only accept latent capsules as the input? Since in your experiments, the number of latent tokens are just 2, I’m curious how two tokens can compress all information about the plan?  For example, the output of the weak decoder in page 9 at least contains information about “Peter”, “Christina”, “number of word”, “misspelling”; are they are stored in two tokens? Or does the weak decoder also accept the problem description as the input?

---

### Official Review · Reviewer_pr2a · 2025-11-01

**Soundness:** 2
**Presentation:** 2
**Contribution:** 2
**Rating:** 2
**Confidence:** 4

**Summary:**

This paper introduces the "Reasoning Capsule" (R-Capsule), a framework to improve the efficiency of CoT reasoning. The core idea is to compress the high-level plan into a small set of latent tokens (the "capsule") which then conditions the generation of explicit execution steps. This method is grounded in the Information Bottleneck principle, using a structural bottleneck to enforce minimality (compression) and a dual-loss (task accuracy + plan reconstruction) to ensure sufficiency. Experiments on math and commonsense benchmarks show R-Capsule improves both accuracy and efficiency (fewer tokens, lower latency) over strong CoT baselines.

**Strengths:**

1. The paper is well motivated. It studies the practical problem of balancing the efficiency and performance of chain-of-thought reasoning.
2. The proposed method is theoretically grounded.
3. Strong empirical results across datasets and model sizes.
4. Extensive ablation studies on the main design choices.

**Weaknesses:**

Despite the strengths, the paper has several significant weaknesses, primarily centered on reproducibility, generalizability, and the rigidity of the proposed framework.

1.The paper lacks crucial details about its novel architectural components, making the work very difficult to reproduce.
2. All experiments appear to be conducted on in-distribution test sets. It is unclear how the learned latent "plan" representations would generalize to out-of-distribution data, such as problems from a different domain, with different phrasing, or of greater complexity than those seen during training.
3. The method assumes a rigid, hierarchical plan -> step structure for reasoning. This limits the method's applicability, as this structure may not be suitable for all types of reasoning problems. This rigid structure also imposes a significant and non-trivial data annotation burden. The framework requires a large dataset of (plan, steps, answer) tuples. The paper uses a large proprietary model (gpt-o3) to generate this data, but the practical challenges, costs, and failure rates of this data generation pipeline are not discussed. This is a major practical barrier to scaling the method to new tasks or domains.
4. The paper includes a model scaling analysis (Fig 2a), which is commendable. However, it is missing a data scaling analysis. Given the high cost of the specialized training data, a crucial analysis would be to show how performance scales with the amount of training data. Does R-Capsule still outperform baselines in a low-data regime (e.g., with 10% or 25% of the data)? This analysis is essential for understanding the method's data efficiency and practical utility.

**Questions:**

1. Could you please provide the precise architectural details and hyperparameters for the "bottleneck network" and the "shallow decoder" (e.g., hidden dimension, layers, attention heads)?
2. Can you comment on the robustness and cost of the data generation pipeline? What was the approximate failure rate of gpt-o3 in generating correct and logically sound (plan, steps) pairs, and how much manual filtering was required?
3. Have you conducted any experiments on out-of-distribution generalization?
4. How do you see the R-Capsule framework being applied to reasoning tasks that do not have a clear, pre-defined plan -> step structure?

---

### Official Review · Reviewer_rPh4 · 2025-11-01

**Soundness:** 2
**Presentation:** 3
**Contribution:** 2
**Rating:** 4
**Confidence:** 4

**Summary:**

This paper introduces the "Reasoning Capsule" (R-Capsule), a novel framework designed to improve the efficiency and accuracy of large language models (LLMs) in complex reasoning tasks. The core idea is to address the high latency and verbosity of standard Chain-of-Thought (CoT) prompting by decoupling the reasoning process into a high-level plan and low-level execution steps. Instead of generating an explicit textual plan, the model learns to compress it into a small set of latent tokens—the Reasoning Capsule.

The method is theoretically grounded in the Information Bottleneck (IB) principle. The capsule is encouraged to be minimal through a low-capacity architectural bottleneck and sufficient through a dual training objective. This objective combines a primary task loss (for answer accuracy) with an auxiliary plan-reconstruction loss, where a separate, shallow decoder is trained to reconstruct the original textual plan from the capsule. This reconstruction loss grounds the latent representation, making it more interpretable and preventing the model from learning uninformative shortcuts.

Experiments on mathematical and commonsense reasoning benchmarks (GSM8K, StrategyQA, etc.) with various model sizes (from GPT-2 to 8B models) show that R-Capsule consistently outperforms standard CoT fine-tuning and other baselines in accuracy, while significantly reducing the number of generated tokens and inference latency.

**Strengths:**

The central contribution—selectively compressing the high-level plan while leaving the execution steps explicit—is both novel and significant. While prior work has explored latent reasoning or hierarchical planning, this paper's core hypothesis that the plan is the right component to compress is a fresh and compelling perspective. It offers a practical solution to the well-known trade-off between the performance of explicit CoT and the efficiency of implicit reasoning. The findings could influence how future models are designed for efficient and structured reasoning.

**Weaknesses:**

1. The experimental section lacks performance evaluation for each model without any training. Including the performance of different prompt strategies (Standard QA, CoT).
2. It is necessary to clarify which version of qwen-3 is being used, whether it is the "instruct" or "thinking" version. The experimental part needs to supplement the performance comparison of different versions of qwen-3.
3. The experimental section requires a more complete comparison, such as the training results of qwen-3-8b on Standard SFT.
4. The method proposed by the author is very similar to the paper "Guiding Language Model Reasoning with Planning Tokens", and the author needs to make a comparison with this method in the experimental part. And analyze what the differences are in the methods.
5. The dataset used by the author is of relatively low difficulty. It is recommended to use some datasets of higher difficulty for evaluation. For example, MATH, HotpotQA, etc.

**Questions:**

Please refer to Weaknesses.

---

### Official Review · Reviewer_x2YQ · 2025-11-01

**Soundness:** 2
**Presentation:** 2
**Contribution:** 2
**Rating:** 4
**Confidence:** 4

**Summary:**

This paper proposes R-Capsule, a framework that compresses the high-level plan of a reasoning chain into a small number of learned latent tokens, while leaving execution lightweight or explicit. The design is motivated by an Information Bottleneck objective: a low-capacity projection enforces minimality, and a plan-reconstruction loss encourages sufficiency and a semantically grounded latent (via a shallow decoder). Experiments on GSM8K, MultiArith, AQuA, StrategyQA, and CSQA2 with small/medium base models (e.g., GPT-2 ~150M, Llama-3-1B, and Qwen3-8B; limited results for 7B/8B) show modest accuracy gains over CoT-SFT and reduced token counts/latency.

**Strengths:**

- Clarity: The paper is generally well written and easy to follow, with a clean separation of planning vs. execution and an intuitive training diagram.
- Core idea: Compressing plans rather than steps is an interesting and plausible hypothesis about where “information” resides in CoT (supported by ablations indicating plan-only compression is best among tested options).
- Methodological framing: The Information Bottleneck view (minimality via a bottleneck; sufficiency via plan reconstruction) is a neat, principled story that helps explain design choices.
- Efficiency gains: The approach reduces visible tokens and latency compared to explicit Plan-CoT on GSM8K (Qwen3), which is practically valuable.

**Weaknesses:**

1.	Questionable premise / under-specified concept of “high-level plans”.

The paper repeatedly leans on “high-level plan” without a precise operational definition or an objective test of plan quality/faithfulness. Reasoning is often non-linear and flexible (re-planning, reflection, backtracking), not strictly “plan-then-execute.” The work assumes a fixed latent plan is the right anchor, yet provides limited analysis of cases where plans need to be revised mid-trajectory.

2.	Experimental scope is limited and misses key settings.

- Model regime: Main results focus on base or very small models (150M–1B) with a small excursion to Qwen3-8B/Llama-3-7B. Today’s “long-CoT” capable models (e.g., o1/R1-style or strong distillations) are absent, making it hard to judge competitiveness in the regime where planning helps most.
- Scaling: The largest model is 8B; there is no 30B+ evidence. The scalability claim therefore remains weakly supported. Figure 2a suggests only small gains at 7B/8B and does not explore larger scales.
- Datasets: Benchmarks are comparatively easy/standard (GSM8K, MultiArith, AQuA, StrategyQA/CSQA2). Stronger contemporary math sets (AIME’24/’25, MATH500, HMMT’25, etc.) are not included, which limits the external validity of the efficiency–accuracy trade-off.
- Baselines: Missing comparisons to models specifically trained for long CoT or modern open distillations (e.g., DeepSeek-distill-Qwen variants), leaving open whether R-Capsule improves over competitive long-reasoning approaches.
- Data generation confound: Training CoT/plan data are synthesized with GPT-o3. The paper does not disentangle how much of the observed improvement comes simply from exposing the base model to high-quality o3 trajectories versus the proposed latent-plan mechanism.

3.	Effect size concerns.

On Qwen3-8B, the reported improvements over Plan-SFT/CoT-SFT appear small (Figure 2a), raising questions about whether the extra complexity (capsule projection + weak decoder + auxiliary loss) is justified at scale.

4.	Technical under-specification.

The paper does not give a rigorous, task-agnostic definition of “plan,” nor a measurable criterion for when a latent token “faithfully” encodes a plan. The link between the abstract plan concept and a fixed number of latent tokens (e.g., K=2) seems heuristic, and the benefits vs. simply using short explicit plans are not thoroughly isolated.

**Questions:**

1.	Definition & validation of “high-level plan.”

- How exactly do you define a high-level plan independently of the training signal?
- Can you provide automatic plan-quality metrics (e.g., plan edit distance vs. reference; causal interventions showing that altering the capsule alters the decoded plan and the solution in predictable ways)?

2.	Re-planning and flexibility.

- Can the model revise its capsule mid-generation when encountering contradictions or errors?
- Have you tried a two-stage or iterative capsule (plan → steps → re-capsule → revised steps) to emulate human re-planning? Any empirical evidence that a fixed K=2 capsule is sufficient?

3.	Stronger baselines and larger models.

- Please compare against recent long-CoT / reflective baselines and strong distillations (e.g., DeepSeek-distill-Qwen, R1-style training).
- Provide results on ≥30B models to substantiate scalability claims. What happens at 32B/70B? Do gains widen or vanish?

4.	Data generation ablations (GPT-o3).

- If you only fine-tune the base model on o3-generated reasoning (no capsule machinery), how much of the gain remains?
- Conversely, if you control for the total o3 tokens seen, does the latent-plan path still outperform explicit short-plan SFT?

5.	Ablations on harder benchmarks and token budgets.

- Report accuracy vs. token-budget curves on AIME’24/’25, MATH500, HMMT’25. Does plan-only compression still dominate?
- Provide per-problem error analysis where capsules help vs. hurt (e.g., multi-turn algebra vs. combinatorics).

6.	Capsule interpretability and diagnostics.

- Beyond the weak-decoder visualization, can you probe capsules with linear classifiers to predict plan attributes (number of subgoals, operation types), or perform targeted perturbations that yield controlled changes in decoded plans?

---

### Note · Authors · 2026-01-09

I have read and agree with the venue's withdrawal policy on behalf of myself and my co-authors.